# STR-MATCH: MATCHING SPATIOTEMPORAL RELEVANCE SCORE FOR TRAINING-FREE VIDEO EDITING

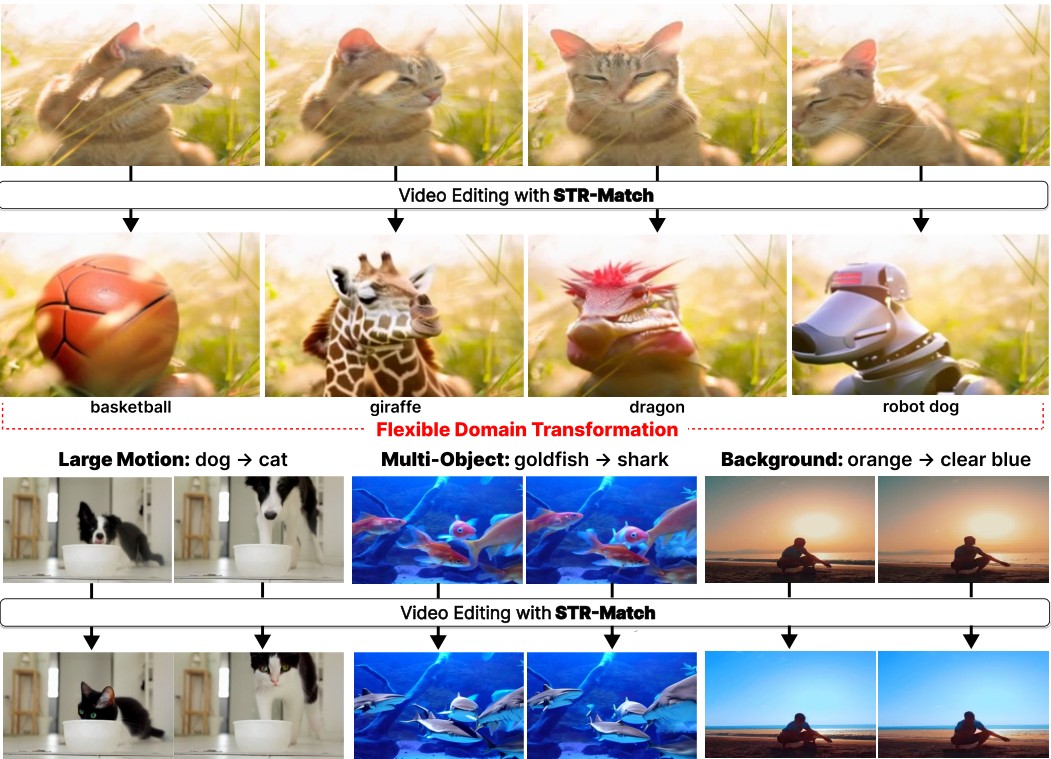

Figure 1: **Generated videos using our proposed algorithm, STR-Match.** Our proposed algorithm, STR-Match, successfully performs flexible domain transformations while preserving the visual information of the source video during the video editing process. It is also applicable to various scenarios, including large motion, multi-object, and background editing.

## ABSTRACT

Existing text-guided video editing methods often suffer from temporal inconsistency, motion distortion, and cross-domain transformation error. We attribute these limitations to insufficient modeling of spatiotemporal pixel relevance during the editing process. To address this, we propose STR-Match, a training-free video editing technique that produces visually appealing and temporally coherent videos through latent optimization guided by our novel STR score. The proposed score captures spatiotemporal pixel relevance across adjacent frames by leveraging 2D spatial attention and 1D temporal attention maps in text-to-video (T2V) diffusion models, without the overhead of computationally expensive full 3D attention. Integrated into a latent optimization framework with a latent mask, STR-Match generates high-fidelity videos with strong spatiotemporal consistency, preserving key visual attributes of the source video while remaining robust under significant domain shifts. Our extensive experiments demonstrate that STR-Match consistently outperforms existing methods in both visual quality and spatiotemporal consistency.

# 1 INTRODUCTION

Diffusion models (Ho et al., 2020; Song et al., 2021a;b) are the leading framework for high-fidelity image and video generation using text prompts. Their applications now extend to tasks such as text-guided image and video editing, where the goal is to generate outputs aligned with target text prompts while preserving regions consistent with both the source and target prompts in the original content. Text-guided image editing typically generates a target image using information extracted during the forward or reconstruction pass of the source image—most commonly via latent optimization (Parmar et al., 2023; Lee et al., 2025) or attention injection (Cao et al., 2023; Tumanyan et al., 2023; Hertz et al., 2023), although some methods adopt alternative strategies such as text embedding interpolation (Kawar et al., 2023; Lee et al., 2024).

While text-guided image editing methods have demonstrated impressive editing capabilities, directly applying them to video editing presents several challenges, including frame inconsistency and undesired motion change. To achieve strong video editing performance while addressing these issues, many prior works (Qi et al., 2023; Jeong & Ye, 2024; Cong et al., 2024; Yang et al., 2025a) leverage pretrained text-to-image (T2I) models augmented with additional components. Some other recent works (Meral et al., 2024; Yatim et al., 2024; Zhang et al., 2025b; Bai et al., 2024) adopt text-to-video (T2V) models to tackle these problems. However, these methods still suffer from the same issues and exhibit degraded performance in challenging scenarios (*e.g.,* large domain shifts).

These limitations in text-guided video editing stem from inadequate modeling of spatiotemporal pixel relevance, which is crucial for producing natural and coherent video content. To address these challenges, we introduce STR-Match, a training-free algorithm that generates videos via latent optimization guided by a novel STR score. The STR score, defined as the multiplicative combination of self- and temporal-attention maps, captures spatiotemporal pixel relevance across adjacent frames by combining 2D spatial and 1D temporal attention from a text-to-video (T2V) diffusion model, without relying on costly full 3D attention maps. This joint formulation offers greater flexibility than treating the attention components separately, as it relaxes excessive constraints and facilitates finding optimal solutions for video editing. Integrated into a latent optimization framework with a masking strategy, STR-Match produces temporally consistent, high-fidelity outputs, effectively handling challenging editing cases and maintaining the key visual attributes of the source.

Our primary contributions are summarized as follows:

- We introduce STR-Match, a novel training-free text-guided video editing approach built upon pretrained T2V diffusion models. It matches spatiotemporal information in the generation process (target latents) to that of the forward process (source latents) via latent optimization, optionally incorporating a latent masking strategy for improved preservation of source content. This design addresses key limitations of existing methods stemming from insufficient modeling of spatiotemporal pixel relevances.

- To obtain spatiotemporal information, we propose the STR score, a spatiotemporal pixel relevance score that combines self- and temporal-attention maps without requiring full 3D attention maps. The STR score also enables flexible optimization, resulting in enhanced overall video quality.

- Through extensive experiments on various video editing tasks, we demonstrate that STR-Match outperforms existing training-free video editing approaches both quantitatively and qualitatively. STR-Match generates temporally coherent, high-fidelity videos with flexible domain transformations, while preserving the visual integrity of the source video. It consistently outperforms prior methods in these aspects.

# 2 RELATED WORKS

## 2.1 TEXT-TO-VIDEO DIFFUSION MODEL

Recent works (Chen et al., 2024; Wang et al., 2024) build on diffusion models by extending pretrained text-to-image (T2I) architectures. These methods commonly introduce lightweight 1D temporal modules into 2D spatial backbones, enabling efficient video generation while preserving the visual priors learned from T2I models. While previous T2V models such as VideoCrafter2 (Chen et al., 2024)

and LaVie (Wang et al., 2024) extend pretrained T2I architectures by inserting lightweight temporal modules into 2D spatial backbones, more recent approaches aim to capture richer spatiotemporal pixel relevances through full 3D attention. Building on advances in efficient attention computation frameworks such as xFormers (Lefaudeux et al., 2022) and FlashAttention (Dao et al., 2022), the latest T2V models (Yang et al., 2025b; Peng et al., 2025) incorporate full 3D attention into their architectures. For example, CogVideoX (Yang et al., 2025b) and Open-Sora-2.0 (Peng et al., 2025) adopt 3D autoencoding architectures with integrated full 3D attention, leveraging FlashAttention to enable efficient attention computation. These methods typically aim to optimize the computation of attention outputs while avoiding the explicit construction of full 3D attention maps, which are computationally expensive and thus impractical to use directly.

## 2.2 TRAINING-FREE VIDEO EDITING METHODS

**T2I-based video editing methods**  With the rapid progress of image editing works (Cao et al., 2023; Hertz et al., 2023; Tumanyan et al., 2023; Parmar et al., 2023; Lee et al., 2025; Si et al., 2025; Kawar et al., 2023; Lee et al., 2024), recent works (Qi et al., 2023; Jeong & Ye, 2024; Cong et al., 2024; Yang et al., 2025a) leverage pretrained T2I models with addtional components to complement frame consistency. FateZero (Qi et al., 2023) manipulates attention maps using binary masks from cross-attention and improves temporal consistency by warping middle-frame features during diffusion. Ground-A-Video (Jeong & Ye, 2024) leverages external models—such as GLIGEN (Li et al., 2023a), RAFT (Teed & Deng, 2020), ZoeDepth (Bhat et al., 2023), and ControlNet (Zhang et al., 2023)—to guide attention modulation with attention maps. FLATTEN (Cong et al., 2024) manipulates attention maps to follow patch trajectories derived from optical flow (Teed & Deng, 2020), aiming to maintain frame consistency. VideoGrain (Yang et al., 2025a) modulates both self- and cross-attention to address multi-grain video editing tasks, relying on external methods (Tumanyan et al., 2023; Cong et al., 2024) to enhance frame consistency. Although these T2I-based methods have demonstrated strong editing capabilities, they still struggle from temporal inconsistency and motion distortion. Moreover, many of these approaches rely on attention injection, which can disrupt the computational graph of the pretrained model and often lead to visual artifacts.

**T2V-based video editing methods**  In contrast to T2I-based approaches, several recent methods (Meral et al., 2024; Yatim et al., 2024; Zhang et al., 2025b; Bai et al., 2024) leverage pretrained T2V models to address temporal consistency in the video editing task. For example, DMT (Yatim et al., 2024) utilizes a pretrained T2V model and introduces a feature descriptor extracted from intermediate layers to guide latent optimization for motion preservation. MotionFlow (Meral et al., 2024) incorporates losses from cross-, self-, and temporal-attention, along with mask-based manipulation, to preserve motion information in the source video. Zhang et al. (Zhang et al., 2025b) extracts motion patterns using temporal modules and applies a frame-to-frame consistency loss during generation. These approaches utilize latent optimization, which preserves the pretrained model's computation graph, allowing for smoother outputs with fewer visual artifacts. However, these methods primarily focus only on motion guidance, which often leads to modifications in unwanted regions (*e.g.*, backgrounds). While UniEdit (Bai et al., 2024) uses attention injection for appearance and motion editing, the foreground and background often become visually decoupled, resulting in incoherent video outputs.

## 3 PRELIMINARY

**Text-to-video diffusion model** We summarize the basic concept of pretrained text-to-video diffusion models as we use the models to perform text-guided video editing. The key components of text-to-video model are an encoder $\text{Enc}(\cdot)$, a decoder $\text{Dec}(\cdot)$, and a noise prediction network $\epsilon_\theta(\cdot)$. Encoder spatially and temporally compresses a video vector $\mathbf{x} \in \mathbb{R}^{F \times H \times W \times 3}$ to a latent vector $\mathbf{z}_0 \in \mathbb{R}^{f \times h \times w \times c}$, and decoder decompresses the latent vector to the video vector. The noise prediction network learns the distribution of latent vectors $\mathbf{z}_0$, and is trained to minimize following objective function:

$$\mathbb{E}_{\mathbf{z}_0, \mathbf{c}, t, \epsilon}[||\epsilon_\theta(\mathbf{z}_t, t, \mathbf{c}) - \epsilon||_2^2], \tag{1}$$

where $\mathbf{z}_0$ denotes the video latent, $\mathbf{c}$ is the corresponding text prompt, $t$ is the diffusion timestep, and $\mathbf{z}_t = \alpha_t \mathbf{z}_0 + \sigma_t \epsilon$ for $\epsilon \sim \mathcal{N}(0, \mathrm{I})$. $\alpha_t$ and $\sigma_t$ are predifined constants satisfying $\alpha_0 = 1$, $\sigma_0 = 0$, and $\sigma_T / \alpha_T \gg 1$.

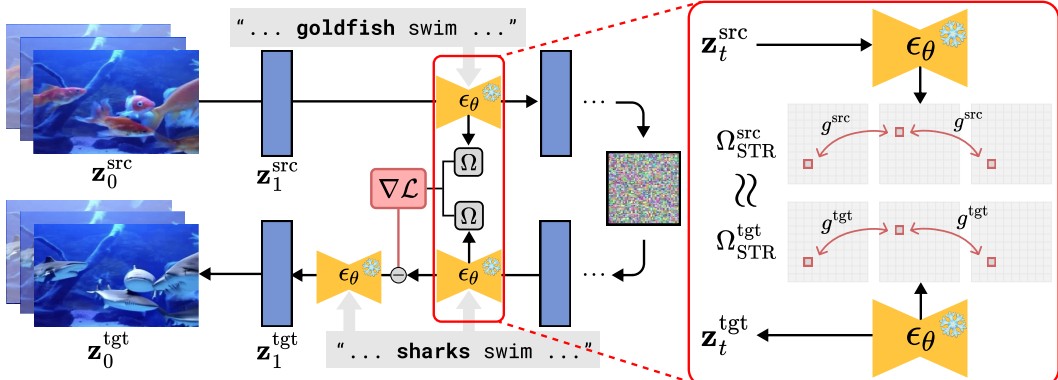

Figure 2: **Illustration of overall STR-Match framework.** We first perform a forward diffusion process, and extract the STR score $\Omega_{\text{STR},t}^{\text{src}}$ from the source video. Then, the target latent is initialized as $\mathbf{z}_T^{\text{tgt}} = \mathbf{z}_T^{\text{src}}$, and during the generation process, we extract the target STR score $\Omega_{\text{STR},t}^{\text{tgt}}$ and optimize the latent $\mathbf{z}_t^{\text{tgt}}$ using a negative cosine similarity between the source and target STR scores. To further preserve unediting regions, we optionally apply a latent mask strategy using a binary mask $M$.

**Attention maps** We specifically focus on two key components of text-to-video models: the spatial self-attention map and the temporal-attention map. Spatial self-attention map, whose dimension is $\mathbb{R}^{f \times h \times n \times n}$, captures relevances between pixels within each frame, where $f$ denotes the number of frames, $n$ represents the number of pixels per frame, and $h$ indicates the number of attention heads. For the rest of the paper, we denote $p, q \in \{1, 2, ...n\}$ for the spatial location of pixel and $i, j \in \{1, 2, ...f\}$ for the frame number. Combining these, $I_i(p)$ represents the pixel at location $p$ in $i$-th frame. Then, the self-attention map element $\text{Attn}(I_i(p) \to I_i(q))$ can be interpreted as importance of $I_i(q)$ to $I_i(p)$ in 2D spatial space. Similarly, temporal-attention map, whose dimension is $\mathbb{R}^{n \times h \times f \times f}$, encodes inter-frame relevances for each pixel, and the element $\text{Attn}(I_i(p) \to I_j(p))$ represents the importance of $I_j(p)$ to $I_i(p)$ in 1D temporal space.

## 4 METHODS

Many text-guided image editing methods (Cao et al., 2023; Hertz et al., 2023; Tumanyan et al., 2023; Parmar et al., 2023; Si et al., 2025) manipulate attention maps, demonstrating that modeling pixel relevances is crucial for effective image editing. Likewise, we expect that spatiotemporal pixel relevances in videos are essential for effective video editing. To this end, we propose the STR score, which captures spatiotemporal pixel relevance across frames by leveraging self- and temporal-attention maps from both U-Net and DiT-based T2V models. It is an aggregation of bidirectional pixel relevances across adjacent frames, efficiently capturing spatiotemporal information and enabling the extraction of key visual attributes from the source video. By integrating the STR score into a latent optimization framework, as illustrated in Figure 2, we enable video editing that preserves source content while achieving high visual quality with flexible domain shifts.

### 4.1 STR SCORE: SPATIOTEMPORAL RELEVANCE SCORE

**U-Net based T2V Models** To quantitatively represent relevance between two pixels $I_i(p)$ and $I_j(q)$ in spatiotemporal space, we define two functions: bidirectional relevance $g(\cdot, \cdot)$, and directional relevance $g(\cdot \to \cdot)$. The directional relevance $g(I_i(p) \to I_j(q))$ quantifies the importance of $I_j(q)$ to $I_i(p)$ in spatiotemporal space, and intuitively, it is expected to be large if both the importance of $I_j(p)$ to $I_i(p)$ and $I_j(q)$ to $I_j(p)$ are high, or the importance of $I_i(q)$ to $I_i(p)$ and $I_j(q)$ to $I_i(q)$ are high. From this motivation, we define directional relevance between $I_j(q)$ given $I_i(p)$ as

$$g(I_i(p) \to I_j(q)) := \text{Attn}(I_i(p) \to I_j(p)) \, \text{Attn}(I_j(p) \to I_j(q))$$
$$+ \text{Attn}(I_i(p) \to I_i(q)) \, \text{Attn}(I_i(q) \to I_j(q)), \qquad (2)$$

for $\text{Attn}(\cdot \to \cdot)$ defined in Section 3. The bidirectional relevance $g(I_i(p), I_j(q))$ extends the directional relevance by considering the connection between $I_i(p)$ and $I_j(q)$ in both directions, as illustrated in Figure 3. Specifically, it is defined as a sum of the importance of $I_j(q)$ to $I_i(p)$ and the

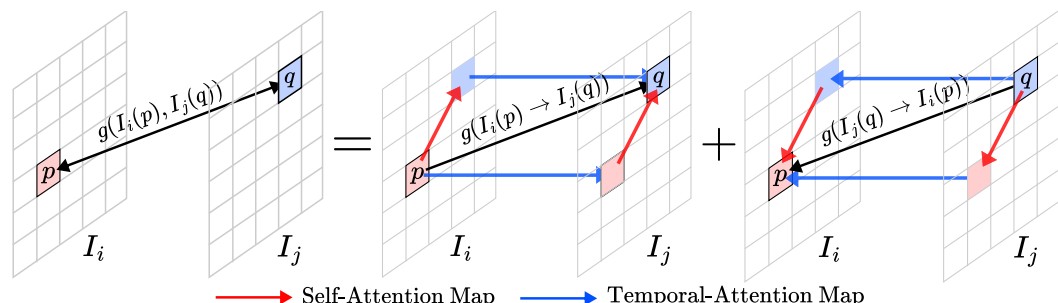

Figure 3: **Illustration of STR score.** (Left) The bidirectional pixel relevance in the spatiotemporal space $g(I_i(p), I_j(q))$ is computed by summing two directional relevance scores along opposite directions. (Right) Each figure illustrates the directional pixel relevance, $g(I_i(p) \rightarrow I_j(q))$ and $g(I_j(q) \rightarrow I_i(p))$, both of which are computed solely through pixel-wise multiplication of self- and temporal-attention maps.

importance of $I_i(p)$ to $I_j(q)$:

$$g(I_i(p), I_j(q)) := g(I_i(p) \rightarrow I_j(q)) + g(I_j(q) \rightarrow I_i(p)). \tag{3}$$

Notably, the bidirectional relevance is fully computed from self- and temporal-attention maps without requiring any additional training or models.

To capture spatiotemporal information in the source video—such as motion and structural layout—we aggregate bidirectional pixel relevances across adjacent frames into a unified representation, termed the STR score. The STR score $\Omega_{\text{STR}}$, or spatiotemporal pixel relevance, is formally defined as follows:

$$\Omega_{\text{STR}}(i, p, q) = \sum_{j \in \mathcal{N}(i)} g(I_i(p), I_j(q)), \tag{4}$$

where $\mathcal{N}(i)$ is a set of neighboring frame numbers to the $i$-th frame.

**DiT-based T2V Models** Recent DiT-based T2V models often bypass the explicit computation of attention maps, which makes it challenging to obtain spatiotemporal relevances, $\text{Attn}(I_i(p) \rightarrow I_j(p))$ and $\text{Attn}(I_j(p) \rightarrow I_j(q))$. To address this limitation, we introduce pseudo self-attention and temporal-attention maps that efficiently approximate these relevances. The query and key matrices generally have the shape $(h, f \times n, c)$, where $c$ denotes the channel dimension and the remaining symbols follow the notation in Section 3. For computation, we reshape the query and key tensors to two forms—$(f \times h, n, c)$ and $(n \times h, f, c)$—to compute pseudo self- and temporal-attention maps via dot-product and softmax. This yields a pseudo self-attention map of shape $(f, h, n, n)$ and a pseudo temporal-attention map of shape $(n, h, f, f)$, which are directly employed in Equation (2). Our formulation enables the efficient computation of STR scores across diverse T2V models, independent of their architectures, without the need to construct full 3D attention maps—a strategy further analyzed in terms of computational complexity in Appendix B.

### 4.2 OVERALL FRAMEWORK: STR-MATCH

The overall procedure of our method is illustrated in Figure 2 and Algorithm 1. We first solve the forward diffusion process of the source video. During the forward process, we extract STR scores $\Omega_{\text{STR},t}^{\text{src}}$ at every timestep and noisy latent $\mathbf{z}_T^{\text{src}}$. Then, starting from $\mathbf{z}_T^{\text{tgt}} = \mathbf{z}_T^{\text{src}}$ as initial point, we perform generation process with latent optimization. For each denoising step, we first optimize the latent variable $\mathbf{z}_t^{\text{tgt}}$, and then solve diffusion process with the optimized latents. The optimization is performed with the following equation:

$$\mathbf{z}_t^{\text{tgt}} \leftarrow \mathbf{z}_t^{\text{tgt}} - \lambda \nabla_{\mathbf{z}_t^{\text{tgt}}} \mathcal{L}_{cos}(\Omega_{\text{STR},t}^{\text{src}}, \Omega_{\text{STR},t}^{\text{tgt}}), \tag{5}$$

where $\mathcal{L}_{cos}$ is a negative cosine similarity, and $\lambda$ is a hyperparameter for controlling the guidance strength. The equation is designed to maximize the cosine similarity between the source and target STR scores, encouraging the spatiotemporal pixel relevances in the target video to align with those of source video to promote preservation of spatiotemporal information.

---

**Algorithm 1** STR-Match

---

1: **Input**: $\mathbf{z}_0^{\text{src}}$ (source video), $p^{\text{src}}$ (source prompt embedding), $p^{\text{tgt}}$ (target prompt embedding), $\Phi(\cdot)$ (ODE solver), $M$ (foreground binary mask, optional)
2: **Hyperparameter**: $\lambda$ (coefficient of negative cosine similarity)
3: **for** $t = 0$ **to** $T - 1$ **do**
4:      $\epsilon_t^{\text{src}} \leftarrow \epsilon_\theta(\mathbf{z}_t^{\text{src}}, t, p^{\text{src}})$
5:      Compute and save $\Omega_{\text{STR},t}^{\text{src}}$ from $\epsilon_\theta(\cdot)$
6:      $\mathbf{z}_{t+1}^{\text{src}} \leftarrow \Phi(\mathbf{z}_t^{\text{src}}, \epsilon_t^{\text{src}}, t \rightarrow t+1)$
7: **end for**
8: $\mathbf{z}_T^{\text{tgt}} \leftarrow \mathbf{z}_T^{\text{src}}$
9: **for** $t = T$ **to** $1$ **do**
10:      Compute $\Omega_{\text{STR},t}^{\text{tgt}}$ from $\epsilon_\theta(\mathbf{z}_t^{\text{tgt}}, t, [p^{\text{tgt}}; p^{\text{src}}])$
11:      $\mathbf{z}_t^{\text{tgt}} \leftarrow \mathbf{z}_t^{\text{tgt}} - \lambda \nabla_{\mathbf{z}_t^{\text{tgt}}} \mathcal{L}_{cos}(\Omega_{\text{STR},t}^{\text{src}}, \Omega_{\text{STR},t}^{\text{tgt}})$
12:      $\epsilon_t^{\text{tgt}} \leftarrow \epsilon_\theta(\mathbf{z}_t^{\text{tgt}}, t, [p^{\text{tgt}}; p^{\text{src}}])$
13:      $\mathbf{z}_{t-1}^{\text{tgt}} \leftarrow \Phi(\mathbf{z}_t^{\text{tgt}}, \epsilon_t^{\text{tgt}}, t \rightarrow t-1)$
14:      **if** $M$ *exists* **then**
15:          $\mathbf{z}_{t-1}^{\text{tgt}} \leftarrow (1 - \texttt{dilate}(M)) \odot \mathbf{z}_{t-1}^{\text{src}} + \texttt{dilate}(M) \odot \mathbf{z}_{t-1}^{\text{tgt}}$
16:      **end if**
17: **end for**
18: **Result**: $\mathbf{z}_0^{\text{tgt}}$ (target video)

---

Since the optimization process preserves the computational graph of the pretrained model, it enables the generation of smooth, high-quality videos while maintaining key visual information from the source. Moreover, since $\Omega_{\text{STR}}$ is conceptually defined as the element-wise product of self- and temporal-attention maps, it enables more flexible optimization compared to using them independently, thereby further enhancing video quality.

**Latent mask strategy** To better preserve regions that are not intended to be edited (*e.g.*, backgrounds), we mix the optimized latent with the latent obtained during the forward process at the same timestep. For a binary mask $M$, where values are $1$ for regions to be edited and $0$ otherwise, and latents $\mathbf{z}_t^{\text{src}}$ obtained during the forward diffusion process of source video, the final target latents are updated as

$$\mathbf{z}_t^{\text{tgt}} \leftarrow (1 - \texttt{dilate}(M)) \odot \mathbf{z}_t^{\text{src}} + \texttt{dilate}(M) \odot \mathbf{z}_t^{\text{tgt}}. \tag{6}$$

This masking strategy ensures to preserve non-target regions in the source video during editing. The latent binary mask is resized and dilated version of segmentation map of editing region of the source video. The `dilate` function is applied to help flexible shape modification.

## 5 EXPERIMENTS

### 5.1 IMPLEMENTATION DETAILS

For U-Net based experiments, STR-Match is implemented using LaVie (Wang et al., 2024) as the pretrained T2V model. For efficient inference, we extract the STR score based on self- and temporal-attention maps, excluding those from the finest resolution. We compare our method against recent training-free video editing algorithms: FateZero (Qi et al., 2023), Ground-A-Video (GAV) (Jeong & Ye, 2024), FLATTEN (Cong et al., 2024), VideoGrain (Yang et al., 2025a), DMT (Yatim et al., 2024), and UniEdit (Bai et al., 2024). For T2I-based methods (FateZero, Ground-A-Video, FLATTEN, VideoGrain), we follow their official implementations. For T2V-based baselines (DMT, UniEdit), we adopt LaVie (Wang et al., 2024) as the pretrained T2V model to ensure a fair comparison. We employ a video segmentation model SAM-Track (Cheng et al., 2023) to obtain binary mask $M$, and OWL-ViT (Minderer et al., 2022), an object detection model to obtain bounding boxes for Ground-A-Video.

For DiT-based experiments, we implement STR-Match using the pretrained CogVideoX-2B (Yang et al., 2025b). To improve computational efficiency during optimization, STR scores are computed using only the first two attention blocks of CogVideoX-2B. We compare our approach with CogVideoX-V2V, a recent training-free video editing method built on the same backbone.

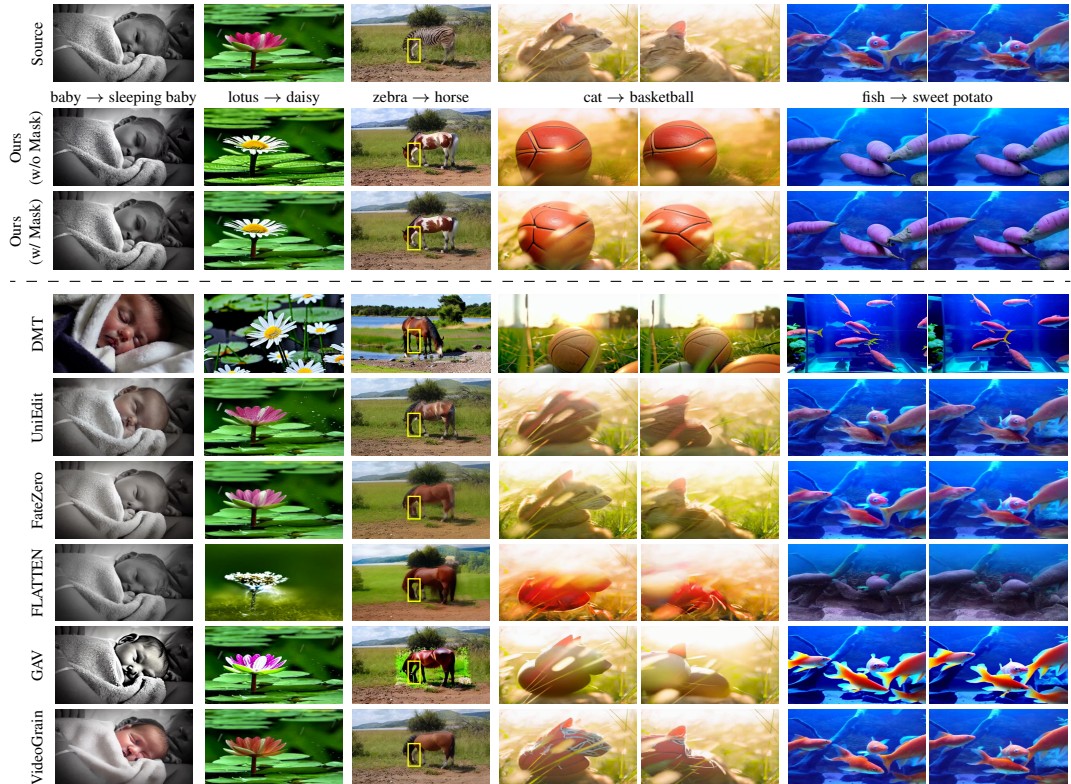

Figure 4: **Qualitative comparisons between STR-Match with LaVie and existing methods.** In each example, STR-Match demonstrates stronger foreground–background texture alignment, higher visual fidelity, better motion alignment, and more flexible shape transformation compared to recent existing methods. Please refer to our HTML-based supplementary material for the edited video results.

All experiments are conducted on a single NVIDIA L40S GPU with 48 GB of memory. We use a fixed hyperparameter $\lambda = 0.01$ and optimize with SGD. For extreme qualitative scenarios (e.g., cat → basketball) in the U-Net based setting, we select $\lambda$ from the range $[0.005, 0.015]$. We set the number of diffusion timesteps to 50 and apply classifier-free guidance (Ho & Salimans, 2021) with a scale of 7.5. A detailed description of the pretrained base models and external models used in implementation is provided in Table 1 of Appendix A.1. Additionally, we provide the results of object deletion/addition and the ablation studies of STR-Match in Appendix D and Appendix E, respectively.

**Quantitative evaluation protocol** For quantitative evaluation in the U-Net based setting, we collect a total of 54 videos, each consisting of 16 frames, comprising samples from the TGVE dataset (Wu et al., 2023) and additional videos sourced from the Internet [1]. We utilize VideoLLaMA3-7B (Zhang et al., 2025a), a pretrained video captioning model, to obtain concise prompts of source videos automatically, and randomly change nouns to construct the corresponding target prompt. We measure four metrics to evaluate the fidelity and fatihfulness of the edited videos to source video and target prompt. Frame Consistency (FC) suggested in VBench (Huang et al., 2024) measures the smoothness of videos, leveraging motion priors in the frame interpolation model (Li et al., 2023b). CLIP Similarity (CS) computes the average CLIP score (Hessel et al., 2021) between the target prompt and edited video. BG-LPIPS (BL) calculates the Learned Perceptual Image Patch Similarity (LPIPS) score (Zhang et al., 2018) between maksed frames of the source and generated videos, where the mask is 1 for regions to preserve. Motion Error (ME) quantifies the average motion difference between the source and generated videos. It is calculated as the pixel-wise differences of optical flows between each video pair, where the optical flows are obtained using RAFT-Large (Teed & Deng, 2020).

For the DiT-based setting, we use 31 synthesized videos, each containing 17 frames. We compare STR-Match against CogVideoX-V2V using the VE-Bench score (Sun et al., 2025), a comprehensive

---
[1]https://www.pexels.com

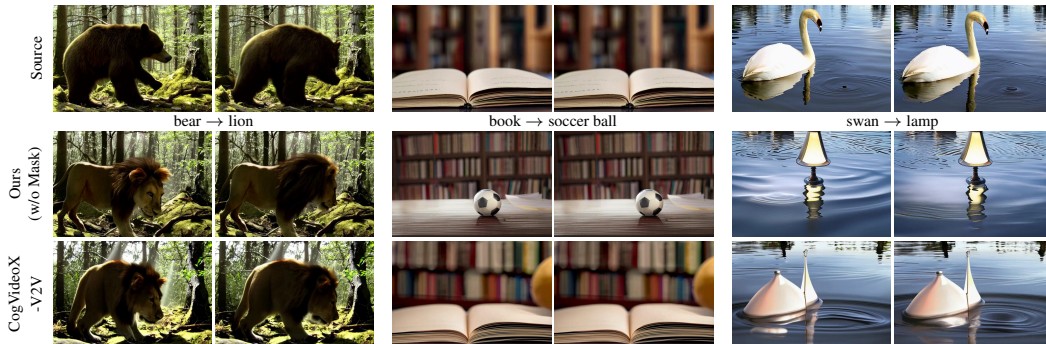

Figure 5: **Qualitative comparisons between STR-Match with CogVideoX and CogVideoX-V2V.** STR-Match demonstrates strong editing capability even in DiT-based settings, outperforming CogVideoX-V2V. In each example, STR-Match performs flexible and effective domain transformation, generating target objects that are natural and faithful to the prompt. In contrast, CogVideoX-V2V either fails to replace the source object or produces results anchored to the source object's shape, resulting in less realistic outcomes.

metric that assesses video editing quality across multiple dimensions, including frame consistency, text alignment, and fidelity to the source video. In addition, we report VE-Bench results in the U-Net based setting to further validate the effectiveness of our approach.

## 5.2 QUALITATIVE RESULTS

**Key features of STR-Match** Figure 1 demonstrates STR-Match's robust editing performance, highlighting its flexibility in challenging scenarios—such as transforming objects into entirely different categories, handling large motion, performing multi-object editing, and modifying background. For instance, transforming a cat into a basketball or a giraffe demonstrates STR-Match's ability to faithfully adapt object shapes to target prompts without being overly anchored to the original shape. Moreover, changing a cat into a dragon or a robot dog—objects unlikely to appear in the original scene—illustrates STR-Match's effective integration of edited elements with the background. These examples emphasize how STR-Match manages domain-shifted objects and significant shape changes, while ensuring the edited elements blend naturally with the background. This combination of flexibility, visual quality, and motion preservation makes STR-Match a powerful tool for diverse video editing tasks.

**Comparison to other editing methods** Figure 4 compares STR-Match with LaVie to recent video editing baselines, showing that our method achieves sharper visual fidelity, tighter foreground–background texture alignment, and more faithful shape transformations. In the 'baby → sleeping baby' case, DMT, UniEdit, and VideoGrain tint the infant while leaving the background gray, whereas STR-Match maintains consistent tonality across the entire frame by capturing spatiotemporal pixel relevance through the STR score. In the 'lotus → daisy' example, several baselines either fail to replace the lotus at all or succeed only by unintentionally changing the background. On the other hand, STR-Match successfully replaces the lotus with high fidelity while preserving the background intact. The same trend holds on more dynamic contents. In the 'zebra → horse' example, most prior methods either fail to capture the horse's leg motion (*e.g.*, lifting its leg) or degrade appearance quality, while Ground-A-Video further disrupts scene consistency. In contrast, STR-Match faithfully reproduces the motion with high visual fidelity.

Furthermore, STR-Match demonstrates strong performance even in extreme video editing scenarios. In the 'cat → basketball' example, most existing methods fail to transform the cat into a basketball, while DMT generates a basketball at the cost of undesired background changes. Similarly, in the 'fish → sweet potato' case, DMT and FLATTEN partially modify the object but suffer from background distortion or low fidelity, and other methods fail to perform the edit. In contrast, STR-Match successfully transforms the object with high visual fidelity while preserving the background. In summary, STR-Match enables high-fidelity, and flexible shape transformation in video editing while preserving spatiotemporal information.

Figure 5 presents a qualitative comparison between STR-Match and CogVideoX-V2V in the DiT-based setting. In the 'bear → lion' example, CogVideoX-V2V generates a lion that remains anchored to the shape of the bear, resulting in an unnatural appearance. In contrast, STR-Match produces a lion with a more natural shape and pose. In the 'book → soccer ball' and 'swan → lamp' cases, CogVideoX-V2V either fails to apply the transformation or yields low-fidelity results, whereas STR-Match successfully performs high-fidelity object transformations. These results demonstrate that STR-Match effectively edits source videos even under the DiT-based setting.

## 5.3 QUANTITATIVE COMPARISON

We quantitatively evaluate STR-Match with LaVie against existing training-free video editing methods for four metrics: temporal consistency (FC), fidelity to the target prompt (CS), background preservation (BL), and motion preservation from the source video (ME). STR-Match, with and without binary masks, achieves strong performance, as evidenced by its large area in the radar graph shown in Figure 6. Notably, compared to T2I-based editing methods, STR-Match demonstrates superior frame consistency, indicating that the proposed STR score effectively captures spatiotemporal pixel relevances from the T2V model.

Furthermore, when comparing STR-Match with masks to UniEdit (red solid and orange lines), both of which utilize SAM-Track, STR-Match outperforms in all evaluated metrics. In the comparison between STR-Match without masks and DMT (red dashed and green lines), the scores reveal that STR-Match more effectively captures key information from the source video, such as background and motion, while maintaining comparable fidelity. This suggests that the STR score achieves a goldilocks balance—preserving essential details from the source video while maintaining the flexibility required for high-fidelity editing—unlike methods that either over-preserve, reducing fidelity, or under-preserve, diminishing faithfulness.

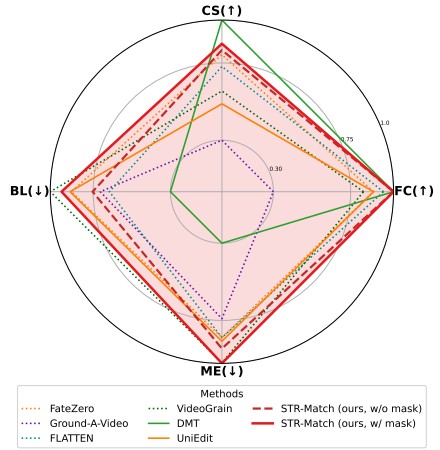

Figure 6: Quantitative comparison between STR-Match with LaVie and existing methods. The solid red line is STR-Match with the binary mask, and the dashed red line is STR-Match without binary mask. The solid lines are T2V-based editing methods, while dotted lines are T2I-based methods. We provide exact metric numbers and analysis in Table 1 of Appendix A.1.

We further evaluate its performance in both U-Net and DiT-based settings using VE-Bench. STR-Match achieves the highest VE-Bench scores across all compared methods in both settings. Detailed results are provided in Table 2 of Appendix A.1.

## 6 CONCLUSION

In this work, we propose a novel spatiotemporal modeling approach that relates to key limitations in existing video editing methods—such as frame inconsistency, motion distortion, visual artifacts, and notably, limited performance in challenging settings like large-gap domain shifts. To overcome these challenges, we propose the STR score, a spatiotemporal pixel relevance score that captures essential video attributes. Notably, it is computed solely from the self- and temporal-attention maps of a pretrained text-to-video (T2V) diffusion model, requiring no additional training or external models. By integrating the STR score into a latent optimization framework alongside a latent mask strategy, we introduce STR-Match, a zero-shot, training-free video editing algorithm that is compatible with any T2V model. Extensive experiments show that STR-Match consistently outperforms existing training-free methods across all quantitative metrics. Moreover, it generates videos with substantially improved visual quality, supporting realistic and flexible domain transformation, preserved motion dynamics, and strong temporal consistency. These results demonstrate both the effectiveness and generalizability of STR-Match, establishing it as a new state-of-the-art baseline for training-free text-guided video editing.

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
