# OpenReview forum: "STR-Match: Matching SpatioTemporal Relevance Score for Training-Free Video Editing"
_ICLR.cc/2026/Conference — ICLR 2026 Conference Withdrawn Submission_

### Official Review · Reviewer_ZKRM · 2025-10-28

**Soundness:** 2
**Presentation:** 2
**Contribution:** 2
**Rating:** 2
**Confidence:** 5

**Summary:**

This paper proposes a STR-Match, a training-free method for text-guided video editing. The core of the method is a SpatioTemporal Relevance (STR) score, which aims to model pixel relationships across frames to maintain temporal consistency and visual fidelity. The STR score is computed by combining self-attention and temporal-attention maps from a pre-trained text-to-video (T2V) diffusion model, avoiding the need for computationally expensive full 3D attention. The authors demonstrate results on both U-Net and DiT-based T2V models (LaVie and CogVideoX).

**Strengths:**

1. A training-free method for text-guided video editing.
2. A SpatioTemporal Relevance (STR) score is proposed to model pixel relationships across frames.

**Weaknesses:**

1. The quality and writing of this paper are inferior. In fig. 1, fig.2, and fig.4, the square video samples from the TGVE dataset are forced to be distorted into rectangles. The appendix link is also broken. There is no appendix in the paper at all. Why did the author add it? Therefore, I think this paper is very hasty and cannot meet the submission standards.
2. The core idea of using attention maps from a pre-trained model to guide editing is well-established in image editing (e.g., Prompt-to-Prompt) and has been extended to video (e.g., FateZero, MotionFlow). The paper does not sufficiently articulate why previous attention-manipulation techniques are Inadequate and why this specific, complex score was necessary to solve the problem.
3. The mathematical formulation of the STR score (Equations 2-4) lacks a clear, intuitive explanation. The authors provide a procedural definition but fail to justify why this specific multiplicative and additive combination of attention maps is the optimal way to capture spatiotemporal relevance. The design feels arbitrary and overly engineered.
4. The paper repeatedly emphasizes avoiding "computationally expensive full 3D attention" as a key advantage. However, calculating the STR score involves combining and optimizing multiple attention maps, and its actual computational overhead and memory usage have not been compared with existing methods. Without training, its practicality and scalability remain questionable.

**Questions:**

Please see Weaknesses.

---

> ### Author Response · Authors · 2025-11-12
>
> Thank you for your valuable feedback. We would like to clarify a few misunderstandings. Our anonymous supplementary HTML page is fully functional (none of the other reviewers reported any accessibility issues), and our supplementary materials include the appendix section, which is provided separately from the main paper. Several reviewers have successfully accessed these materials and provided comments on them.
>
> Regarding the visualization issue, we intentionally resized the video samples into a rectangular format because our T2V models operate at rectangular resolutions. The displayed videos are the direct outputs of our method, and we explicitly noted this resolution detail in our supplementary page.
>
> We hope this clarification resolves the misunderstanding, and we kindly ask you to reconsider your evaluation of our submission in light of these facts.

---

### Official Review · Reviewer_eFQs · 2025-10-30

**Soundness:** 3
**Presentation:** 3
**Contribution:** 3
**Rating:** 4
**Confidence:** 4

**Summary:**

This paper presents STR-Match, a training-free method for consistent text-driven video editing.
The key idea is to compute spatiotemporal pixel relevance score and integrate it into a latent optimization (like guidance).
They show that their results are temporal coherence, maintaining semantic alignment with the edited prompt and preserving source appearance.

**Strengths:**

1. Strong results – the qualitative examples (in the supplementary HTML and ZIP files) are impressive. Even for non-trivial edits involving large domain shifts, STR-Match maintains high fidelity and coherence.

2. Clarity and presentation – the paper is well organized and easy to follow, with intuitive figures (2,3) which make the method readable and clearer.

**Weaknesses:**

1. Dependence on the number of temporal neighbors - the method appears to rely heavily on the number of frames (neighbors) used when computing the spatiotemporal relevance. This parameter directly affects qualitative quality, quantitative scores, and runtime. However, it is not reported or ablated (as far as I could find). Clarifying how many neighbors are used, and analyzing the method’s sensitivity to this choice is important.

Dataset curation - the paper states that the authors collected their evaluation videos. Most of the shown examples involve slow or smooth motion, which may favor methods based on feature similarity. It is unclear whether STR-Match would perform as well on complex or fast motions (e.g., parkour, running, or hand-held camera movement). Please comment on the diversity of your dataset and whether results generalize to such settings.

Scalability to longer videos - the presented results seem limited to short clips (~16–17 frames).
How does the method scale to longer sequences, where relevance estimation and temporal accumulation could drift over time?

Minor presentation issue - (1) the teaser figure (cat example) appears stretched — the aspect ratio seems incorrect, which slightly reduces the professional appearance of the paper. (2) Figure 2: Should z_1​ actually be z_t​? The figure seems to illustrate an intermediate denoising step rather than the first latent.

**Questions:**

My questions are already mentioned in the Weaknesses section

---

### Official Review · Reviewer_T6dk · 2025-11-02

**Soundness:** 2
**Presentation:** 2
**Contribution:** 2
**Rating:** 4
**Confidence:** 4

**Summary:**

The paper presents score distillation-based, training free, text-guided video editing framework, STR-Match. In addition to existing sds based editing framewokr, STR-Match proposes to extract spatial and temporal attention maps, and applies pixel-relevance guidance in the latent optimization stage. The paper experiments with both U-Net based and DiT based T2V models, outperforming existing methods.

**Strengths:**

- The paper proposes a v2v editing method that is model-training free.
- The paper conducts extensive comparison with sota baselines and show superiority against them.
- The paper conducts comprehensive ablation studies.

**Weaknesses:**

- The paper is based on DDS (Delta denoising Score, ICCV 2023) framework, and extends the score-based text-guided editing to t2v setup. Thus, naive extension of DDS on t2v models should be ablated.
- The paper has very close philosophy as DreamMotion (ECCV 2024), which is also a training-free, score distillation-based video editing method. The work also exploits decomposed spatial and temporal attention maps for their guidance term. There should be a discussion on how STR-match is different from DreamMotion and also compared with the method.
- The paper lacks a computational overhead comparison with baselines, which is an important analysis for training-free video editing methods.
- Important ablation studies should be included in the main paper.

**Questions:**

- What is the reference for the CogVideoX-V2V method?
- Although the method is model training free, it requires latent optimization. How much more computational overhead is needed for the latent optimization stage, in both terms of time and memory?
- Is it necessary to apply the loss $L_{cos}$ in every denoising step?
- What is the reason/ground for choosing the first two blocks of CogVideoX DiT model for extracting attention maps?
- Is Eq.(5) applied single time (single time solver) for each denoising step? Or does it require multiple optimizations within a denoising step?

---

### Official Review · Reviewer_NTVJ · 2025-11-06

**Soundness:** 3
**Presentation:** 3
**Contribution:** 3
**Rating:** 6
**Confidence:** 4

**Summary:**

This paper proposes training-free video editing approach by matching spatiotemporal attention feature between source and target latents.
To obtain spatiotemporal information, we propose the STR score, a spatiotemporal pixel relevance score that combines self- and temporal-attention maps. The effectiveness of proposed approach has been verfied with several base T2V diffusion models in multiple benchmarks.

**Strengths:**

Belows are strong points that this paper has:

1. The paper is clear and enjoyable to read, providing a straightforward explanation of the motivation and methodology.

2. The proposed STR score matching between source and target is a novel yet simple approach that can be easily applied to any pre-trained text-to-video (T2V) model.

3. The experiments are thoughtfully designed and effectively demonstrate the strength and validity of the proposed method.

**Weaknesses:**

1. Although the authors present computational complexity in Appendix B, details regarding inference time and additional FLOPs should be included to enable a more comprehensive comparison.

2. The current STR score combines spatial and temporal attention components. To better demonstrate the complementary effects between these two aspects, it would be helpful for the authors to include an ablation study—such as using the full STR score, spatial-only, and temporal-only variants.

**Questions:**

Please check above listed in Weaknesses section.

---

### Note · Authors · 2025-11-14

I have read and agree with the venue's withdrawal policy on behalf of myself and my co-authors.